# Hypokalemia in Diabetes Mellitus Setting

**DOI:** 10.3390/medicina58030431

**Published:** 2022-03-16

**Authors:** Lucas Coregliano-Ring, Kleber Goia-Nishide, Érika Bevilaqua Rangel

**Affiliations:** 1Department of Medicine, Nephrology Division, Federal University of São Paulo, São Paulo 04038-901, Brazil; lucas.ring@unifesp.br (L.C.-R.); kleber.nishide@unifesp.br (K.G.-N.); 2Instituto Israelita de Ensino e Pesquisa, Albert Einstein Hospital, São Paulo, São Paulo 05652-900, Brazil

**Keywords:** diabetes mellitus, hypokalemia, kidney and heart disease

## Abstract

Diabetes mellitus is a public health problem that affects millions of people worldwide regardless of age, sex, and ethnicity. Electrolyte disturbances may occur as a consequence of disease progression or its treatment, in particular potassium disorders. The prevalence of hypokalemia in diabetic individuals over 55 years of age is up to 1.2%. In patients with acute complications of diabetes, such as diabetic ketoacidosis, this prevalence is even higher. Potassium disorders, either hypokalemia or hyperkalemia, have been associated with increased all-cause mortality in diabetic individuals, especially in those with associated comorbidities, such as heart failure and chronic kidney disease. In this article, we discuss the main conditions for the onset of hypokalemia in diabetic individuals, briefly review the pathophysiology of acute complications of diabetes mellitus and their association with hypokalemia, the main signs, symptoms, and laboratory parameters for the diagnosis of hypokalemia, and the management of one of the most common electrolyte disturbances in clinical practice.

## 1. Introduction

According to the World Health Organization, diabetes mellitus (DM) contributes to 11.3% of deaths globally and an estimated 4.2 million deaths among 20–79-year-old adults are attributable to that chronic condition [1]. DM-attributed deaths possess regional disparity, ranging from 6.8% (lowest) in Africa to 16.2% (highest) in the Middle East and North Africa. About half (46.2%) of the deaths attributable to DM occur in people under the age of 60 years. Africa has the highest (73.1%) proportion of deaths attributable to DM in people under the age of 60 years, while Europe has the lowest (31.4%) [1].

Hypokalemia is one of the most common electrolyte disturbances in clinical practice and is usually secondary to poor glycemic control associated with polydipsia/polyuria, in particular diabetic ketoacidosis (DKA) and hyperglycemic hyperosmolar state (HHS), gastro-intestinal loss combined with hypomagnesemia, and diuretic use for controlling edema in chronic kidney disease (CKD) or heart failure (HF) due to cardio-renal syndrome.

A higher risk of atrial fibrillation, respiratory muscle impairment, Q-T interval increase, torsade des pointes, and ventricular fibrillation, and, ultimately, higher morbidity and mortality in diabetic individuals with HF and CKD are clinical conditions associated with hypokalemia [2]. Therefore, there is an association between baseline serum K^+^ levels and mortality in these patients (Figure 1). All three conditions (CKD, HF, and DM) can be associated with higher mortality rates over an 18-month follow-up when compared with control individuals. All-cause mortality was also higher at the extremes, both in K^+^ values below 4.0 mEq/L and in values above 6.0 mEq/L [2].

## 2. Hypokalemia and DM

### 2.1. Definition, Prevalence, and Importance

Hypokalemia is defined as a plasmatic potassium (K^+^) concentration < 3.5 mEq/L. The prevalence of hypokalemia in individuals over 55 years of age and with DM, varies between 1.0 and 1.2%. This prevalence is even higher in individuals with associated CKD and increases with age [3]. One of the main risk factors associated with hypokalemia is the use of diuretics: 10 to 50% of patients being treated with this type of medication can develop hypokalemia [3,4]. Other medications that contribute to hypokalemia in individuals with diabetes are insulin, beta-2 agonists (through activation of the sodium-potassium ATPase [Na^+^-K^+^-ATPase] pump), antiarrhythmic agents, glucocorticoids and mineralocorticoids, antibiotics (penicillin and aminoglycosides), antifungals (amphotericin B), and excessive use of laxatives.

There is also an association between hypokalemia and the risk of developing diabetes in prediabetic individuals [5]. There is evidence of an inverse relationship between serum K^+^ level and the risk of developing incidental DM, regardless of diuretic use. One study found hazard ratios of 1.2–1.3 for the development of incidental DM in subjects with serum potassium K^+^ below 5.0 mEq/L when compared to subjects whose potassium levels were above this range [6]. Hypokalemia induced by the use of thiazide diuretics was also associated with an increased risk of developing DM. However, early-stage control of Thiazide-induced hypokalemia may correct glucose intolerance to prevent the development of diabetes in these individuals [7]. Despite plasma insulin and serum glucose increase and serum K^+^ decreases in hypertensive patients under hydrochlorothiazide monotherapy, no changes of serum glucose were noted whether serum K^+^ levels were greater than 4.0 mEq/L or below 4.0 mEq/L [8].

The relation of hypokalemia and hyperglycemia is based on the function of the ATP-sensitive potassium (K_ATP_) channel in islet cells [9]. Thus, when glucose enters ß-cell through the GLUT2 transporter it is metabolized to glucose-6-phosphate. The increase in intracellular adenine nucleotides inhibits K_ATP_ and leads to its closure. The subsequent membrane depolarization activates the voltage-dependent calcium (Ca^2+^) channels, which triggers insulin exocytosis after the increase in Ca^2+^ intracellular. Thus, hypokalemia is associated with the development of hyperglycemia through the impairment of potassium-dependent insulin release in response to glucose overload.

In healthy individuals, mild hypokalemia may be completely asymptomatic and pose little or no risk whatsoever. However, in individuals with DM, especially in those with cardiovascular comorbidities, mild to moderate hypokalemia may pose a high risk of morbidity and mortality.

### 2.2. Main Causes of Hypokalemia in Individuals with DM

In individuals with DM, hypokalemia results from three different groups of events: transcellular shifts (particularly due to high dosages of insulin or as a result of metabolic acidosis), abnormal losses (gastrointestinal losses, renal losses, dialysis, or hypomagnesemia), or inappropriate intake [2,3,4,10]. The main cause of hypokalemia in individuals with DM is the use of high doses of insulin, whether during the treatment of type 1 DM (T1DM) and advanced stages of type 2 (T2DM) or while attempting to correct acute events such as DKA and HHS [11,12]. Another risk factor that is strongly associated with hypokalemia in diabetic individuals is the use of diuretics, especially thiazides and loop diuretics. In Table 1, we document the main causes of hypokalemia according to the mechanism.

#### 2.2.1. Transcellular Shifts Caused by Drugs

##### Insulin

The physiological response to insulin is the activation of the sodium-potassium ATPase (Na^+^-K^+^-ATPase) pump, promoting the rapid entry of K^+^ into peripheral cells. In addition, the insulin-induced passage of glucose into cells is a fuel source to maintain the action of the Na^+^-K^+^-ATPase pump, providing feedback for this K^+^ transport mechanism. At normal levels, insulin promotes only a transient reduction in serum K^+^ concentration, which will be normalized by the gradual release of K^+^ back into the plasma. However, high doses of insulin, whether due to incorrect administration during the treatment of T1DM and T2DM in advanced stages or during the treatment of acute complications of DM, can lead to hypokalemia [13], which is the most common cause of low serum K^+^ concentrations in individuals with DM [4].

##### Beta-2 Agonists

Sympathomimetic Beta-2 agents are another important group of drugs that promote the activation of the Na^+^-K^+^-ATPase pump. The main examples of Beta-2 agonists that can induce hypokalemia are drugs used to treat asthma, such as antispasmodic agents and bronchodilators (albuterol, terbutaline, ephedrine, metaproterenol, isoproterenol, fenoterol, pirbuterol), decongestants (pseudoephedrine), tocolytics (ritodrine and nylidrin), dopamine, and HF treatment (especially dobutamine). Hypokalemia induced by these agents can persist for several hours and can reach levels as low as 2.5 mEq/L depending on dosage and route of administration. Although theophylline is not classified as an α–adrenergic agonist, it is an antispasmodic agent that also stimulates Na^+^-K^+^-ATPase activity. Toxic levels of this agent can rapidly induce extreme hypokalemia [4].

##### Other Agents

Therapeutic doses of antiarrhythmic agents, such as verapamil, do not present an increased risk for the development of hypokalemia in diabetic individuals. However, intoxication with high doses of verapamil can induce severe hypokalemia. High doses of chloroquine and barium can also induce hypokalemia by inhibiting the release of K^+^ by cells [4].

#### 2.2.2. Abnormal Potassium Losses Caused by Drugs

##### Diuretics

One of the most common causes of hypokalemia in individuals with DM is the use of diuretics. Through different pathways, both loop diuretics and thiazides increase the Na^+^ supply to the collecting duct. The increase in Na^+^ concentration in this segment induces its reabsorption, creating an electrochemical gradient that favors eliminating potassium. The degree of hypokalemia depends both on the dosage of diuretic used and on the Na^+^ concentration in the distal segments of the nephron. Moreover, the combination of more than one class of diuretics, such as a loop with a thiazide or a thiazide analog, potentiates K^+^ secretion, which may facilitate the onset of hypokalemia. Drug-induced hypokalemia may be associated with both metabolic acidosis and alkalosis, either by retaining bicarbonate (HCO_3_^−^) or by inhibiting the antiporters Na^+^ and hydrogen (H^+^), induced, e.g., by acetazolamide.

Loop diuretics act by inhibiting the sodium-potassium-chloride (Na^+^-K^+^-2Cl^−^) co-transporter located in the thick ascending limb of Henle’s loop [14]. These drugs also inhibit the reabsorption of magnesium (Mg^2+^) and Ca^2+^, by compromising the potential difference between the tubular lumen and the interstitium (the main driving force for the reabsorption of these ions in this segment). The increase in urinary K^+^ excretion after loop diuretic use is due to various mechanisms: (1) The increase in Na^+^ supply, especially to the collecting duct, intensifies the excretion of K^+^ and H^+^ (Na^+^ reabsorption by the main cells creates an electrochemical gradient that enables the secretion of K^+^ into the tubular lumen, through the ROMK [renal outer medullary potassium] channels); (2) Activation of the renin-angiotensin-aldosterone system (RAAS) due to volume depletion and a decrease in sodium chloride (NaCl) transport at the dense macula; (3) Release of vasopressin, in response to Na^+^ and volume depletion. An increase in the urinary excretion of H^+^ and K^+^ can lead to hypochloremic alkalosis states and hypokalemia, especially if an inadequate intake of K^+^ is associated.

Thiazide diuretics inhibit Na^+^-Cl^−^ cotransporter (ENCC1 or TSC) located in the apical membrane of cells in the convoluted distal tubule [14]. The expression of this protein is closely regulated by aldosterone. The proximal tubule may be a secondary target for the action of these drugs. Inhibitors of the Na^+^-Cl^−^ symport increase intratubular K^+^ and H^+^ by the same mechanisms as loop diuretics. In the case of the diuretics (loop diuretics and thiazide diuretics), the increased amount of intra-luminal Na^+^ delivered to the distal segments of the nephrons enters the principal cells via the epithelial Na^+^ channel (ENaC) channel. This creates a negative electrical charge inside the lumen. Activation of Na^+^-K^+^-ATPase causes it to pump Na^+^ into the blood and exchange for K^+^. Drawn by the negative electrical charge in the lumen, K^+^ will leave the cell to enter the lumen via the ROMK channel, leading to hypokalemia. Inside the intercalated cell of the collecting ducts, the stimulation of the H^+^-K^+^-ATPase pump, which along with the negative charge in the collecting duct lumen, causes H^+^ to exit intercalated cells into the lumen. Therefore, alkalosis metabolic develops (serum HCO_3_^−^ concentration, 28–36 mmol/L) and leads to hypokalemia as well [4]. Thiazide diuretics can reduce glucose tolerance and exacerbate latent DM [15]. The mechanism of impaired glucose tolerance apparently involves changes in glucose metabolism and impaired insulin secretion. Thiazide-induced DM, however, does not appear to offer the same cardiovascular risk as incident DM [16].

Administration of K^+^ with the thiazide diuretic can prevent hyperglycemic events. In hypertensive patients, Thiazide-induced hypokalemia may compromise treatment efficacy. Thiazide diuretics can also affect the lipid profile of diabetic patients, raising plasma levels of total cholesterol, low-density lipoprotein, and triglycerides [14]. In hypertensive individuals, chlorthalidone was associated with an increased risk of hypokalemia (hazard ratio [HR], 2.7), CKD (HR, 1.24), acute kidney failure (HR, 1.37), and DM [HR, 1.24] without offering a reduction in the risk of cardiovascular events, when compared with hydrochlorothiazide [17].

##### Glucocorticoids and Mineralocorticoids

Glucocorticoids, such as hydrocortisone, prednisone, and prednisolone do not directly interfere with K^+^ excretion in the kidneys. Importantly, glucocorticoids are usually given at doses that produce minimal mineralocorticoid stimulation (comparing to cortisol, prednisone and prednisolone have 0.8 mineralocorticoid potency, whereas hydrocortisone mineralocorticoid potency is equal to cortisol, fludrocortisone has 125–150 potency mineralocorticoid when compared to cortisol, and dexamethasone does not have mineralocorticoid potency) to avoid the side effects associated with activation of the aldosterone pathway, which leads to hypokalemia, volume expansion, and hypertension [18].

Mineralocorticoids, such as fludrocortisone, can induce K^+^ depletion in the distal nephron, through their action on mineralocorticoid receptors, located in the apical membrane of tubular cells, which stimulate the expression and activity of Na^+^-K^+^-ATPase pump, ENaC, and ROMK channels, inducing Na^+^ reabsorption and K^+^ secretion [4]. Other substances with glucocorticoid action, especially licorice derivatives (Glycyrrhiza glabra), such as carbenoxolone or cottonseed derivatives (gossypol), can also induce hypokalemia due to their inhibitory effects on 11-hydroxysteroid dehydrogenase [19].

##### Antibiotics

Beta-lactam antibiotics, such as penicillin, can lead to renal K^+^ losses, when administered intravascularly and in high doses, by increasing the Na^+^ supply to the distal segments of the nephron. Aminoglycosides and amphotericin B can cause hypokalemia due to disturbances in electrolyte homeostasis. Treatment with amphotericin B can lead to hypokalemia and hypomagnesemia in up to 90% of cases, depending on the dose. More than one mechanism has been related to amphotericin B-induced electrolyte disturbances such as the induction of pore formation in the membrane of renal tubular cells, changes in the H^+^-K^+^-ATPase pump in the distal tubule, leading to renal tubular acidosis (RTA) type I, and increased absorption of Na^+^ in the gastrointestinal tract, with resultant excretion of K^+^ in feces [20]. Aminoglycoside-induced hypokalemia may be associated with hypomagnesemia [21]. Due to their positive charge, aminoglycosides can bind to polyva lent cation receptors in the distal tubule, inhibiting Mg^2+^ reabsorption. Another suggested mechanism for renal K^+^ loss is the stimulation of sodium and chloride channels, leading to hypokalemic metabolic alkalosis [21].

##### Oral Anti-Diabetics and Potassium

In overweight individuals with T2DM, linagliptin (a type 4 dipeptidyl-peptidase inhibitor) increased the renal excretion of Na^+^ and K^+^ when compared to sulfonylurea glimepiride [22]. The use of glycosuric agents (sodium-glucose cotransporter-2 inhibitors), especially empagliflozin and dapagliflozin, is increasingly common in diabetic patients with CKD and HF. These drugs promote an excellent improvement in volume through the increase in urinary output and natriuresis, although they have not shown significant effects on K^+^ wastage [23,24,25].

#### 2.2.3. Induced Gastrointestinal Losses

Despite being often disregarded when investigating hypokalemia, gastrointestinal losses due to excessive use of laxatives or enemas can also lead to K^+^ wastage, and should always be considered during an investigation, especially in individuals with weight loss and dehydration. Prolonged vomiting leads to K^+^ losses, as K^+^ is found on gastric secretions (10 mEq/L), and when volume depletion occurs, RAAS is activated and promotes ultimately K^+^ renal losses [10].

#### 2.2.4. Nondrug-Induced Transcellular Shifts

Acute anabolic states, in which a strong stimulus from growth factors, colony-stimulating factors, and other mediators, promotes cell proliferation, and potassium shifts from the extracellular environment to the interior of cells where formation can occur. Some conditions that can induce hypokalemia in this context are high-grade lymphomas, treatment for anemia due to B12 deficiency, and acute leukemias [4]. Other conditions that can induce severe hypokalemia (<3.0 mmol/L) are: Graves’ disease thyrotoxicosis (hypokalemia occurs due to a rapid and massive shift of K^+^ from the extracellular to the intracellular compartment, as thyroid hormone stimulates Na^+^-K^+^ ATPase transcription, and enhances the activity and membrane of this pump in skeletal muscle cells, which can be associated with signs and symptoms of muscle weakness); primary hyperaldosteronism (due to excess mineralocorticoids); familial hypokalemic paralysis (an autosomal dominant mutation in the gene encoding the dihydropyridine receptor); delirium tremens (related to the high serum concentration of catecholamines), and barium intoxication (due to blockage in the output of K^+^ from the cells) [4].

#### 2.2.5. Nondrug-Induced Potassium Losses

Inadequate K^+^ intake is a very rare cause of hypokalemia when not associated with another condition. For reductions in serum potassium concentration, dietary intake should be below 1 g per day (25 mmol). Considering the high concentration of potassium in the intracellular environment, even in severe starvation scenarios, tissue lysis, especially in the musculature, releases potassium into the plasma, attenuating the potassium depletion [4]. Diarrheal scenarios, in which there is a massive loss of water and electrolytes through the gastrointestinal tract, can result in symptomatic hypokalemia. In diabetic individuals, any condition that increases the predisposition to diarrhea can lead to potassium losses, such as acute and chronic complications of DM, exocrine pancreatic insufficiency, infections, neoplasms, inflammatory bowel diseases, malabsorption syndromes, and others [4,10]. Another common cause of metabolic alkalosis in individuals with DM is excessive Na^+^ reabsorption in the distal nephron, associated with K^+^ spoliation, which may be secondary to mineralocorticoid activity or abnormalities in renal transport. The prevalence of primary aldosteronism in individuals with new-onset T2DM and hypertension is at least 20% [26]. Among individuals with primary hyperaldosteronism, there is evidence that the prevalence of DM is close to 21%, higher than an estimated 12% prevalence of DM in the general population [27]. In type I renal tubular acidosis, due to inadequate H^+^ secretion in the distal tubule, hypokalemia can occur and is usually corrected with the administration of sodium bicarbonate. In cases of type II acidosis, due to inadequate reabsorption of HCO_3_^−^ in the proximal tubule, hypokalemia is uncommon, but it can be aggravated with the administration of HCO_3_^−^ [28]. Hypomagnesemia can also lead to hypokalemia or hinder its treatment, by increasing renal K^+^ excretion [29]. As stated earlier, drugs that induce hypomagnesemia, such as amphotericin B and loop or thiazide diuretics, or conditions such as hyperaldosteronism and diarrhea can lead to hypokalemia, both directly and due to magnesium loss. Other causes of hypomagnesemia include alcoholism, intrinsic renal tubular transport disorders, such as Bartter and Gitelman syndromes, and tubular injuries from other nephrotoxic drugs, in particular aminoglycosides and cisplatin. Hypomagnesemia is very common in hypokalemic individuals with T2DM [30,31]. Metabolic alkalosis is often associated with the development of hypokalemia (hypokalemic metabolic alkalosis). One of the main causes of metabolic alkalosis is vomiting, a situation in which hypokalemia is secondary to chloride depletion and reduced renal K^+^ reabsorption. In this scenario, chloride replacement resolves both alkalosis and hypokalemia. Thus, K^+^ depletion stimulates H^+^ secretion in the collecting duct, through H^+^-K^+^-ATPase and HCO_3_^−^ reabsorption in the ascending limb of the loop of Henle. Potassium depletion also downregulates both Na^+^-K^+^-2Cl^−^ and Na^+^-Cl^−^ cotransporters, increasing Na^+^ delivery to the collecting duct and further stimulating the collecting duct H^+^ secretion. When K^+^ deficiency is severe, this effect results in measurable Cl^−^ excretion despite Cl^−^ depletion and sustained metabolic alkalosis even when sodium chloride is administered. Additionally, K^+^ depletion stimulates ammonium production, facilitating the acid excretion needed to sustain metabolic alkalosis [31,32]. From a cellular point of view, the mechanism by which Mg^2+^ deficiency leads to refractoriness of potassium correction is [32]:

(i) In the late distal tubular and cortical collecting duct cells, Na^+^-K^+^-ATPase in the basolateral membrane pumps K^+^ into the cells and then K^+^ is secreted into the lumen via apical K^+^ channels (ROMK and maxi-K/BK [Big potassium]/Slo1/KCa1.1 channels). To note, ROMK is an inward-rectifying K^+^ channel responsible for basal (not stimulated by flow) K^+^ secretion, so that inward rectification indicates that K^+^ enters the cells faster than K^+^ is pumped out the cells. The Maxi-K channel is a large-conductance calcium-activated K^+^ channel.

(ii) Na^+^ reabsorption via ENaC depolarizes the apical membrane potential, providing the driving force for K^+^ secretion, as the lumen presents a negative charge. Aldosterone increases Na^+^ reabsorption via ENaC to stimulate K^+^ secretion through the ROMK channel and Na^+^-K^+^ ATPase, whereas Maxi-K channels are responsible for flow-stimulated K^+^ secretion. Mg^2+^ binds and blocks the pore of the ROMK channel, which leads to its inward rectification, thereby limiting K^+^ efflux. Potassium influx can displace intracellular Mg^2+^ from the pore and release the block. The concentration of intracellular Mg^2+^ required for inhibiting ROMK relies on the membrane voltage and extracellular concentration of K^+^. At zero intracellular Mg^2+^, K^+^ ions move in or out of the cell through the ROMK channel freely. At intra- (140 mM) and extracellular (5 mM) K^+^ concentrations, the chemical gradients drive K^+^ outwards. When the membrane potential is negative inside the cells, it drives K^+^ inward.

(iii) Inward and outward movement of K^+^ ions reach an equilibrium at −86 mV equilibrium potential [Ek] = −60 × log 140/5). When the membrane potential is more negative than Ek (−100 mV), K^+^ ions move into the cells. On the other hand, at a membrane potential higher than Ek (−50 mV), K^+^ moves out of the cells. At the physiologic Mg^2+^ intracellular concentration (1.0 mM), ROMK inward rectifies K^+^, as K^+^ efflux is blocked when intracellular Mg^2+^ binds to ROMK. The influx of K^+^ displaces intracellular Mg^2+^, allowing maximal K^+^ entry. These ROMK properties mediate K^+^ secretion in the distal nephron, which is regulated by intracellular Mg^2+^. Notably, when inward conductance is greater than outward, K^+^ influx does not occur, as membrane potential is more positive than Ek.

Therefore, at physiological conditions, ROMK is inhibited by the intracellular concentration of Mg^2+^ which ranges from 0.1 to 10.0 mM, with the median at approximately 1.0 mM [26]. Thus, intracellular Mg^2+^ is a critical determinant of ROMK-mediated K^+^ secretion in the distal nephron. Changes in intracellular Mg^2+^ concentration could significantly affect K^+^ secretion. Acute complications of DM can also evolve with hypokalemia, either during crises or during insulin treatment.

#### 2.2.6. Genetic Causes of Hypokalemia

Hypokalemia may also be found in a broad set of genetic disorders. Briefly, from a pathophysiologic perspective, hypokalemia may be divided into disorders with a low urine K^+^ excretion or disorders with a high urine K^+^ excretion, as reviewed elsewhere [33].

In the genetic disorders associated with low urine K^+^ excretion rate, three mechanisms are involved:(a)increased K^+^ shift, including familial hypokalemia periodic paralysis or FPP (an autosomal dominant disorder caused by mutations on ion channel genes encoding the dihydropyridine-sensitive voltage-gated Ca^2+^ channel α1-subunit (*CACNA1S*) [FPP type I] and tetrodotoxin-sensitive voltage-gated Na^+^ channel α-subunit [*SCN4A*] [FPP type II] of skeletal muscle) and Andersen-Tawil syndrome (associated with mutations in the gene (*KCNJ2*) encoding a pore-forming subunit of the inward rectifier K^+^ channel protein, Kir2.1, which is expressed in heart and skeletal muscles);(b)defects in the intestinal tract (characterized by a mutation in the downregulated in adenoma (DRA) gene encoding a Cl^−^-OH^−^ (HCO_3_^−^) exchanger expressed in the apical membranes of the colon and ileum, which leads to watery diarrhea, hypochloremic metabolic acidosis, and hypokalemia);(c)defects in exocrine glands (cystic fibrosis is associated with defective chloride reabsorption by the dysfunctional CFTR [cystic fibrosis transmembrane regulator] in the sweat ducts which is responsible for excessive Cl^−^ and Na^+^ loss in sweat, leading to ECF volume depletion and, ultimately, to secondary hyperaldosteronism). To note, in familial hypokalemia periodic paralysis, attacks can be induced not only by rest after exercise, carbohydrate-rich meals, or exposure to cold but also by the administration of glucose or insulin or glucocorticoid, which can put diabetic patients at greater risk.

In the genetic disorders associated with high urine K^+^ excretion rate, two mechanisms are involved in:(a)increased urine flow rate to cortical collecting ducts;(b)increased K^+^ concentration in the cortical collecting ducts.

An increased urine flow rate to cortical collecting ducts is caused by increased excretion of electrolytes due to diuretic use or when a tubular defect is found or the increase in osmole excretion is due to non-electrolytes, such as mannitol, glucose, or urea. Therefore, a diabetic patient with higher levels of plasma glucose may present hypokalemia which is secondary to an increase in urine flow rate to cortical collecting ducts due to glycosuria.

When increased K^+^ concentration in the cortical collecting ducts is observed, we may speculate that a disorder characterized by fast Na^+^ reabsorption in the cortical collecting ducts is present. The most common disorders found in this setting are genetic hypokalemia associated with mineralocorticoid excess state, including glucocorticoid-remediable aldosteronism (a disorder caused by mutations in 11β-hydroxylase (CYP11B1) gene which encodes the aldosterone synthase gene for aldosterone biosynthesis in the adrenal zona glomerulosa and is regulated by angiotensin II and aldosterone synthase (CYP11B2) which encodes 11β-hydroxylase gene for cortisol biosynthesis in the adrenal zona fasciculate and is regulated by adrenocorticotrophic hormone [ACTH]), congenital adrenal hyperplasia due to 11β-hydroxylase or 17α-hydroxylase deficiencies [11β-OHD and 17α-OHD], Liddle’s syndrome (mutations in both β and γ subunits of ENaC associated with a deletion or alteration in their cytoplasmic C termini, which leads to a lack of its internalization via clathrin-coated pits pathway or its degraded via Nedd4 pathway, and therefore ENaC remains in an activated form on the cell surface), and apparent mineralocorticoid excess (a rare disorder that is caused by mutations in the gene (*HSD11B2*) encoding renal-specific 11β-hydroxysteroid dehydrogenase type 2 (11β-HSD2), which is responsible for converting cortisol to cortisone in the principal cells of distal tubules and crucial for protecting the mineralocorticoid receptor from being occupied by cortisol). In all these disorders, the excess of mineralocorticoid, either by augmented aldosterone secretion or production of other steroids, upregulates ENaC activity and leads to sodium reabsorption and hypertension associated with hypokalemia.

The second group of disorders associated with increased K^+^ concentration in the cortical collecting ducts includes a mechanism in which Cl^−^ reabsorption in this segment is diminished, which includes disorders such as hypochloremic metabolic alkalosis, in particular Bartter´s syndrome (an autosomal recessive renal tubular disorder characterized by defective reabsorption of NaCl in the Henle´s loop, including five subtypes of mutations in genes encoding the Na^+^-K^+^-2Cl^−^ cotransporter [NKCC2], K^+^ channel [ROMK], kidney-specific Cl^−^ channel [CLCNKB], barttin [BSND], and calcium-sensing receptors [CaSR]) and Gitelman´s syndrome (secondary to defective reabsorption of NaCl in the distal convoluted tubule due to mutations in the *SLC12A3* gene, which encodes the thiazide-sensitive Na^+^-Cl^−^ cotransporter [NCC] on the apical membrane of that tubule). In both syndromes, hypokalemia is associated with renal K^+^ and Na^+^ wasting, with low or normal blood pressure and secondary hyperreninemia and hyperaldosteronism. Whereas in Bartter´s syndrome, a high urine Ca^2+^ and Mg^2+^ excretion are found, in Gitelman´s syndrome a low urine Ca^2+^ and high Mg^2+^ excretion are invariably found.

Importantly, Gitelman´s syndrome patients may be at greater risk of developing insulin resistance and type 2 DM [34], as chronic hypokalemia and hypomagnesemia impair insulin secretion and sensitivity, whereas hyperaldosteronism increases insulin resistance. From a molecular and cellular point of view, when glucose enters the pancreatic ß cell via GLUT2 transporter, it is metabolized to glucose-6-phosphate leading to changes in the intracellular concentration of adenine nucleotides that inhibit the K_ATP_ channel and cause its closure. When it occurs, the membrane depolarization activates subsequently the voltage-dependent calcium channels, leading to an increase in intracellular calcium, which triggers insulin exocytosis. Mutations in the ATP-Sensitive Potassium-Channel Subunit Kir6.2 lead to permanent neonatal DM [9].

In addition, hypomagnesemia found in Gitelman´s syndrome patients is implicated in a reduced tyrosine kinase activity at the insulin receptor level, and dysregulates K^+^ -ATP and L-type Ca^2+^ channels in the ß cells, which impairs insulin activity and secretion. Hyperaldosteronism can increase reactive oxygen species, accelerate endothelial remodeling, which can reduce the delivery of insulin for glucose metabolism, and promote insulin resistance by reducing insulin receptor substrate-1 expression, and by blocking the downstream protein kinase B signaling in the vascular smooth muscles [35].

The second cause of increased K^+^ concentration in the cortical collecting ducts comprises hyperchloremic metabolic acidosis, in particular, inherited isolated proximal RTA (renal tubular acidosis associated with defects in the electroneutral Na^+^-H^+^ exchanger 3 [NHE3] in the luminal membrane or the electrogenic Na^+^-HCO_3_^−^ cotransporter [NBC1] in the basolateral membrane, which is associated with a reduction in the reabsorption of filtered HCO_3_^−^), inherited distal RTA (mutation in the genes encoding two H^+^-ATPase subunits specific to intercalated cells, including ATP6V1B1 and ATP6V0A, or mutations in H^+^-K^+^-ATPase and AE1 Cl^−^-HCO_3_^−^ exchanger [SLC4A1], leading to an impaired ability to excrete H^+^ in distal tubules), and inherited proximal and distal RTA (mutation in the gene encoding carbonic anhydrase II which generates H^+^ in distal convoluted tubules) [33].

In proximal RTA, the initial insult is associated with an excess of HCO_3_^−^ excretion and the urine pH is higher than 6.5. However, over time, HCO_3_^−^ serum levels drop and an impaired absorption is again sufficient to acidify urine to lower than 5.5. Thus, proximal HCO_3_^−^ excess loss in the urine causes an increase in urine flow rate to the distal nephron, leading to potassium wasting and activation of RAAS from mild hypovolemia and, therefore, to hypokalemia. Other features associated with RTA include the augment in glucose, uric acid, phosphate, and amino acids in urine (Fanconi syndrome) [33].

In distal RTA, the decreased H^+^ in tubule lumen draws out K^+^ causing hypokalemia. In addition, these patients present calcium phosphate kidney stones caused by decreased citrate excretion and hypercalciuria, as well as by the fact that salts are more likely to precipitate at higher urine pH [33].

### 2.3. DM-Related Acute Complications: Diabetic Ketoacidosis (DKA), Hyperglycemic Hyperosmolar State (HHS), and Euglycemic Diabetic Ketoacidosis (EDKA)

DKA and HHS are the main and potentially fatal acute complications of DM. It was believed that DKA was exclusive to T1DM, with insulin deficiency, and that HHS was exclusive to T2DM, with insulin resistance. However, reports are showing that they can occur in both T1DM and T2DM [13]. DKA is more frequent in individuals with T1DM, being a potentially serious and common condition in hospital emergencies, especially in the pediatric age group, caused by a major deficiency in insulin secretion. Conversely, HHS is more often present in T2DM and is characterized by hyperglycemia, hyperosmolarity, and dehydration in the absence of ketoacidosis. Despite representing less than 1% of hospital admissions, it has a mortality rate 10 times higher than DKA, so HHS is the acute hyperglycemic complication of DM with the highest mortality (5 to 20%) [36]. Mixed cases of DKA with a hyperosmolar component and HHS with ketoacidosis can also occur, in up to one-third of the cases. Both conditions are the extremes of a continuum, in which the manifested signs and symptoms change according to the levels of insulin deficiency, insulin resistance, dehydration, acidosis, and energy metabolism (Table 2).

#### 2.3.1. Diabetic Ketoacidosis

The clinical manifestations of DKA can quickly onset, ranging from a few hours to a couple of days, and include polydipsia, polyuria, nausea, vomiting, abdominal discomfort, and weight loss. The presence of metabolic acidosis can stimulate the medullary breathing center, leading to Kussmaul’s breathing pattern (hyperventilation). Other findings may include ketone breath and signs of dehydration such as altered skin turgor, dry mucous membranes, hypotension, and tachycardia. A reduction in body temperature is common, and in severe cases, hypothermia can onset. If fever is present, the possibility of infection should be investigated. The mental state can range from alertness to profound lethargy. Upon arrival at the hospital, 10% of cases may have a loss of consciousness.

The classical triad of DKA includes hyperglycemia (>11 mmol/L or known diabetic patient), ketonemia (blood level > 3 mmol/L or ketonuria 2+ on dipstick), and acidosis (HCO_3_^−^ < 15 mmol/L and/or pH < 7.3) with high anion gap [13]. The pathophysiology of DKA begins with a relative or absolute insulin deficiency associated with an increase in counterregulatory hormones, especially glucagon, glucocorticoids, and catecholamines. This state of hormonal imbalance leads to an increase in blood glucose and stimulation of hepatic gluconeogenesis, glycogenolysis, and insulin resistance, with low tissue glucose consumption. Glucocorticoids, especially cortisol, increase the supply of amino acids at the expense of stimulating protein catabolism. Glucagon and catecholamines, in turn, induce the activation of the enzyme glycogen phosphorylase, increasing glycogenolysis. Furthermore, an increase in the glucagon/insulin ratio inhibits precursors of the glycolytic pathway, decreasing glycolysis. The depletion of electrolytes (especially potassium) and acidosis also interfere with insulin action in target tissues. Glycosuria secondary to hyperglycemia results in osmotic diuresis, leading to water and electrolytes loss, hypovolemia, dehydration, and reduced GFR, further aggravating the hyperglycemia and creating a vicious cycle [13].

Ketogenesis is also a result of this hormonal imbalance state, with insulin deficiency and excess of counter-regulatory hormones. In this state, hormone-sensitive lipase is activated in the adipose tissue, resulting in free fatty acid (FFA) debt into the blood. In the liver, lipid metabolism is shifted to oxidation of these FFAs, which are transformed in ketone bodies. Glucagon, in turn, inhibits glycolysis at the hepatic level, by reducing the synthesis of malonyl-CoA. This process also deviates the metabolism towards the oxidation of fatty acids in ketone bodies—acetone, acetoacetate, and ß-hydroxybutyrate—and the release of H^+^ ions. The entry of K^+^ in the cells is decreased due to both insulin deficiency and metabolic acidosis. An increase in extracellular osmolarity, as a result of osmotic diuresis and electrolyte translocation, is present in both DKA and HHS, and induces volume outflow from the intra to extracellular space, with consequent cellular dehydration and a small dilution in plasma Na^+^ concentration. Important renal K^+^ losses occur due to osmotic diuresis, with reduced NaCl reabsorption, and ketonuria. Other factors that can also aggravate dehydration and electrolyte imbalance are the use of diuretics, vomiting, diarrhea, and reduced water intake.

Despite the total body K^+^ deficiency ranging between 3–5 mEq/kg in DKA, patients often have elevated plasma K^+^ concentrations, with normokalemia or hypokalemia being important indications of K^+^ deficiency. During insulin treatment, plasma K^+^ levels will invariably fall, possibly leading to hypokalemia [13]. Recent studies have shown that the incidence of hypokalemia in DKA may be lower than 4%, appearing, in most cases, only after insulin administration [37]. In emergency department patients, however, the prevalence of hypokalemia associated with DKA can be as high as 11% [38]. Rhabdomyolysis is another possible complication of DKA, occurring more frequently in cases of prolonged acidosis, high serum creatinine, hyperkalemia, and ketonuria. Rhabdomyolysis per se can also aggravate hyperkalemia, by releasing potassium from the cytoplasm [39].

#### 2.3.2. Hyperglicemic Hyperosmolar State

Usually, HHS has a more insidious onset than DKA, and it may take several days or even weeks for the first symptoms to appear. The main clinical manifestations are polyuria, polydipsia, weakness, and blurred vision. Sensory changes are common in HHS and mental status can range from fully alert to coma. A seizure can occur in up to 20% of patients [36]. Signs of dehydration are reduced turgor, dry mucous membranes, cold extremities, hypotension, and reflex tachycardia [27]. Diagnostic criteria of HHS include plasma glucose 30–33.3 mmol/L (540–600 mg/dL), pH > 7.30, HCO_3_^−^ > 15–18 mmol/L, urine acetoacetate negative or low positive, osmolality 320 mmol/kg and presentation with stupor or coma, severe dehydration, and feeling unwell [13,36].

Similar to DKA, in the HHS there is also hyperglycemia secondary to glycogenolysis, increased gluconeogenesis, and decreased entry of glucose into peripheral tissues. In this case, however, the insulin deficiency is relative. Insulinopenia is also accompanied by an increase in counter-regulatory hormones (glucagon, catecholamines, and glucocorticoids). However, in HHS the concentrations of FFAs, cortisol, and glucagon are lower when compared to those values found in DKA. In this case, it is believed that insulin deficiency is sufficient to compromise the adequate use of glucose by tissues, but insufficient to shift energy metabolism towards lipolysis, ketogenesis, and metabolic acidosis [13]. In HHS, osmotic diuresis due to glycosuria is also present, resulting in loss of water and electrolytes and dehydration [39]. However, as this scenario settles in a longer period of time than in DKA, usually the volume deficit is greater. Hypovolemia also leads to dehydration and decreased GFR, aggravating hyperglycemia. This “snowball effect” leads to higher blood glucose and osmolality values than those found in DKA.

Analogous to diabetic ketoacidosis, the main acute complication of HHS treatment is hypokalemia secondary to insulin administration [36].

#### 2.3.3. Euglycemic Diabetic Ketoacidosis

EDKA is a situation in which blood glucose remains below 200 mg/dL and is associated with alcohol intoxication, pregnancy, prolonged fasting, and depression in patients with T1DM, acute pancreatitis, and salicylate poisoning [13]. In recent years, it has reappeared with the ascension of sodium-glucose cotransporter-2 (SGLT-2) inhibitors for the treatment of DM and cardiovascular diseases. These drugs prevent the renal reabsorption of sodium and glucose in the proximal tubule and increase glucosuria in patients with DM, decreasing glycemia. At the same time, these drugs induce an increase in plasma glucagon. This increase is not able to raise blood glucose concentration due to glycosuria. However, high glucagon induces a reduction in plasma insulin levels (reducing the insulin/glucagon ratio), promotes lipolysis (20% enhanced lipid oxidation), decreases carbohydrate oxidation (falls by 60%), and increases the production of FFAs, substrates for the production of ketones. Although recommended for patients with T2DM only, gliflozins have seen off-label use as an adjunct to insulin therapy for T1DM. Recent studies show EDKA as a complication of SGLT2 inhibitors in both groups of diabetic patients [13].

In addition to the use of SGLT-2 inhibitors, EDKA usually has a triggering event—situations of metabolic stress such as surgery, myocardial infarction, stroke, prolonged fasting, and strenuous exercise. Patients using SGLT-2 inhibitors should avoid alcoholic beverages, ketogenic diets, maintain a good level of hydration and discontinue medication before any stressors, such as surgery. Other precipitating factors of EDKA comprise a reduction in carbohydrate intake, reduction of insulin dose in the context of good glycemic control, cocaine use, and pregnancy [13]. Therefore, K^+^ should be monitored and corrected accordingly.

## 3. Symptoms, Exams, and Diagnosis of Hypokalemia

As the ion K^+^ plays a large role in the physiology of various tissues, organs, and systems, its deficiency can lead to changes in cardiovascular functioning, in skeletal muscles, in the kidneys, and even in the release and effect of certain hormones [10]. The direct correlation between K^+^ levels and the appearance of signs and symptoms is not linear, depending on intrinsic factors and clinical status of each individual, highlighting diabetic patients, in which it may vary according to both K^+^ levels and the presence of other pre-existing comorbidities. Nevertheless, mild hypokalemia can be often asymptomatic [40].

Although chronic or persistent hypokalemia may be asymptomatic in some individuals, patients with DM may have this condition worsened by diarrhea or vomiting, which can occur during acute complications of DM. Nocturia and polyuria can also be exacerbated, especially in individuals predisposed to persistent hypokalemia, as in Bartter and Gitelman syndromes. Hypokalemia-induced polyuria is related to an impairment of vasopressin action in collecting ducts. In addition, insulin treatment can also promote K^+^ shift into cells. Therefore, hypokalemia can have worse consequences in diabetic patients, which puts these individuals at greater risk of chronic hypokalemia. In this group of patients, cardiovascular diseases are found more often, making them more vulnerable to cardiac arrhythmias, fluid depletion, and worsening neuropathy from muscle weakness [41,42]. To note, K_ATP_ channels may not function properly in the DM setting because its expression is reduced in myocardium cells and aortic smooth muscle cells, resulting in impaired heart and vascular function [43]. Consequently, hypokalemia may affect the membrane potential and pose a decreased response to stress conditions, such as hypoxia and oxidative stress. Importantly, as hypokalemia may lead to hyperglycemia due to the impairment of insulin secretion and peripheral glucose utilization, a vicious circle is triggered where hypokalemia worsens glucose control and vice-versa.

### 3.1. Cardiovascular Effects

The main cardiovascular changes caused by hypokalemia are cardiac arrhythmias [10]. Low K^+^ concentration increases cardiac muscle excitability and delays its repolarization, which can induce both atrial and ventricular arrhythmias [44]. The most commonly observed ECG changes are shown in Figure 2, which include T wave flattening, ST-T segment depression, an extension of the QT interval [44], presence of U waves, and multiple ventricular extrasystoles, which can be seen in up to 20% of patients with severe hypokalemia (>2.6 mmol/L) [_bookmark3138]. Patients at greatest risk for developing life-threatening arrhythmias are the elderly or those with underlying ischemic heart disease. Hypertensive patients using hydrochlorothiazide seem to have a higher risk for the incidence of sudden death [10]. The main serious arrhythmias induced by hypokalemia are ventricular fibrillation, ventricular tachycardia, and torsades des pointes.

### 3.2. Muscular Effects

In contrast to cardiac musculature, hypokalemia can induce hyperpolarization of skeletal muscle, compromising its ability to depolarize and contract. Additionally, dehydration (e.g., during diabetic ketoacidosis) can reduce blood supply to the musculature and induce rhabdomyolysis. Together, these processes can lead to muscle weakness and fatigue. In severe cases, hypokalemia can cause respiratory muscles weakness and even lead to respiratory acidosis [44].

### 3.3. Kidney Effects

The most common renal complication of hypokalemia is metabolic alkalosis, which can occur through multiple pathways: The low serum K^+^ concentration promotes H^+^ secretion through the H^+^-K^+^-ATPase pump in the collecting ducts. Furthermore, it stimulates the absorption of HCO_3_^−^ in the proximal tubule, NH4^+^ synthesis, and reduction in urinary citrate secretion. Another effect of hypokalemia in the kidneys is the impairment of the urinary concentration capacity, apparently through defective activation of the enzyme adenylate cyclase in the tubular cells of the distal nephron, preventing the activity of the antidiuretic hormone. In addition, fluid intake is stimulated due to an increase in the level of angiotensin II in the central nervous system. This hypokalemic-induced nephrogenic diabetes insipidus can lead to polyuria, with loss of up to 3 L of water per day. When associated with hyperaldosteronism, hypokalemia can also lead to cystic kidney disease, originating from the collecting duct epithelium [10].

### 3.4. Hormonal Effects

In diabetic patients, the effects of low K^+^ concentration on insulin have great importance. Hypokalemia leads to both a reduction in pancreatic insulin release and its activity in target cells. The combination of these effects can worsen hyperglycemia and diabetic control [44], having devastating effects in individuals in DKA or HHS state.

### 3.5. Diagnosis of Hypokalemia

In the presence of the aforementioned signs and symptoms and after the identification of serum K^+^ < 3 mmol/L, it is important to perform a sequential analysis of the possible causes and mechanisms behind hypokalemia. The first step is to assess any possiblerenal K^+^ losses, differentiating them from possible gastrointestinal losses. Some measurements can be used to identify whether the causes are of renal or extrarenal origin, such as the transtubular potassium gradient (TTKG), the urinary potassium excretion fraction, or the potassium value obtained in an isolated urine sample, which can be normalized by creatinine (K/Cr ratio) [45]. It is important to keep in mind that each of these measurements has its due limitations, for example, not very sensitive to losses due to mineralocorticoid activity. Additionally, because they are fixed values, they can be influenced by other variables, such as volume and electrolyte intake, urinary flow, and GFR. Furthermore, TTKG is more sensitive in detecting inappropriate K^+^ secretion in hyperkalemia [44].

#### 3.5.1. Fractional Excretion of Potassium (*FE_K_*)

*FE_K_* is the percent of K^+^ filtered into the proximal tubule that appears in the urine. For an individual with normal kidney function with an average dietary K^+^ intake, the *FE_K_* is approximately 10%. When hypokalemia is the result of extrarenal causes (low K^+^ intake, increased K^+^ shifts into cells, and gastrointestinal loss), the kidney conserves K^+^ and, consequently, *FE_K_* is low. Conversely, hypokalemia secondary to renal losses is associated with an increase in *FE_K_*. In contrast, in the setting of hyperkalemia, a high *FE_K_* suggests an extrarenal etiology, whereas a low *FE_K_* is consistent with a renal etiology.
(1)$$FE_{K}=\frac{Cl_{K}}{Cl_{Cr}}=\frac{\frac{U_{K}}{S_{K}}}{\frac{U_{Cr}}{S_{Cr}}}\times 100\%$$
(2)$$Cl_{K}=\frac{\left(V\times U_{K}\right)}{S_{K}}$$
(3)$$Cl_{Cr}=\frac{\left(V\times U_{K}\right)}{S_{Cr}}$$*FE_K_*: Fractional excretion of potassium; *Cl_K_*: Clearence of potassium; *Cl_Cr_*: Clearence of creatinine; *U_K_*: Urinary potassium; *S_K_*: Serum potassium; *U_Cr_*: Urinary creatinine; *S_Cr_*: Serum creatinine; V: Urinary volume.

If a urine creatinine measurement is not available, one can often use *U_K_* alone, in a random urine specimen, to differentiate between renal and extrarenal causes of hypokalemia: *U_K_* > 20 mEq/L suggests a renal etiology, whereas *U_K_* < 20 mEq/L suggests extrarenal etiology.

#### 3.5.2. Transtubular Potassium Gradient (TTKG)

The transtubular potassium gradient estimates the potassium gradient between the urine and the blood in the distal nephron. TTKG is a measurement of net K^+^ secretion by the distal nephron, after correcting for changes in urine osmolality. In a normal individual under normal circumstances, the TTKG is about 6 to 12.
(4)$$TTKG=\frac{U_{K}/\frac{U_{Osm}}{P_{Osm}}}{P_{K}}$$*U_K_*: Urinary potassium; *U_Osm_*: Urinary osmolality; *P_Osm_*: Plasma osmolality; *P_K_*: Plasmatic potassium; *U_Cr_*: Urinary Creatinine.

In the hypokalemia setting, a high TTKG suggests excessive renal K^+^ losses, whereas hypokalemia with a low TTKG suggests an extrarenal etiology. The *U_osm_*/*P_osm_* term is included to correct for the rise in *U_K_* that is due purely to water abstraction and concentration of the urine. Several factors limit the utility of the *FE_K_* and TTKG in the differential diagnosis of K^+^ disorders, so that the *FE_K_* and TTKG are increased when K^+^ intake is increased, and they are decreased when K^+^ intake is decreased. In patients with kidney function impairment, there is an adaptive increase in K^+^ excretion per functioning nephron, and *FE_K_* and TTKG may increase accordingly. Figure 3 describes a flowchart to guide the etiological diagnosis of hypokalemia.

### 3.6. Management of Hypokalemia

For the optimal treatment of hypokalemia, it is necessary that underlying causes have already been identified and associated disorders are being managed. Significant potassium losses, for example, due to vomiting, diarrhea, or excessive diuresis, need to be ceased. In most cases, K^+^ disturbances are accompanied by acid-base disturbances and, for this reason, the acid-base status should be constantly monitored [44]. If metabolic acidosis is present, for example, due to diabetic ketoacidosis or type I tubular acidosis, correction of hypokalemia should be performed before administration of bicarbonate. Before starting K^+^ replacement, hypomagnesemia, if present, should be promptly corrected with intravenous administration of magnesium sulfate, as Mg^2+^ deficiency may prevent the correction of hypokalemia [40]. The next step is the administration of K^+^, which can be achieved orally (in liquid or tablet form), or intravenously (KCl solution is the most frequent).

The amount of potassium that should be administered depends on the total K^+^ deficit, which can be calculated based on the serum potassium concentration. A commonly used equation is:(5)$$\;\;K_{deficit}\left(mmol\right)\;=\;\left(K_{lower\;limit*}-K_{measured}\right)\;\times \;Weight\;\left(Kg\right)\;\times \;0.4\;\;\;$$

$K_{deficit}$: Serum potassium deficit (in mmol); $K_{lower\;limit*}$: Serum potassium lower limit under normal conditions; $K_{measured}$: Serum potassium measured concentration; Weight (Kg): Bodyweight (in Kilograms). * Potassium normal lower limit ranges from 3.0 to 3.5 mmol/L.

Without any stimulus for transcellular shifts, a 0.1 mmol/L reduction in K^+^ concentration, on average, equates to a total body deficit of approximately 35 mmol. If the replacement route of choice is intravenous, potassium administration rate should not exceed 20 mmol/h (increases the serum K^+^ by about 0.25 mmol/L), in order to avoid the onset of hyperkalemia, and in cases of associated periodic paralysis hypokalemia, this rate should not exceed 10 mmol/h, due to a spontaneous improvement in these conditions [44].

If faster replacement is required, 20 or 40 mmol/h can be given via a central venous catheter due to the risk of phlebitis if a peripheral vein is cannulated for this purpose. Importantly, continuous ECG monitoring should be used under these circumstances. In DKA and HHS, serum K^+^ can be normal or elevated on admission despite total body K^+^ depletion, which is more severe in HHS compared to DKA (Table 1) [13,36]. Osmotic-induced intracellular dehydration results in K^+^ efflux from the cells. Since insulin causes a shift of K^+^ into the cell, via an indirect effect on Na^+^-K^+^ ATPase, one should correct the K^+^ level to >3.3 mEq/L before starting insulin therapy. In that case, insulin must be held. If K^+^ is between 3.3 and 5.3 mEq/L, 20–30 mEq of K^+^ should be given in each liter of intravenous fluid to keep serum K^+^ between 4 to 5 mEq/L [37]. Potassium should be monitored if >5.3 mEq/L. Magnesium should be checked and given intravenously whether necessary, as this approach is important to prevent renal wasting of K^+^ with exacerbation of hypokalemia. Routine administration of phosphate is not recommended. However, careful phosphate replacement can be considered in patients with very low levels (<1 mEq/L) due to the risk of cardiac dysfunction or respiratory distress [46].

In the DKA setting, major guidelines for K^+^ replacement emphasize the importance of blood gas and renal function tests for profiling replacement [47,48,49,50]. Initial rehabilitation with saline solution is recommended until serum K^+^ levels normalize. Insulin should be withheld if blood K^+^ is below 3.3 mmol/L to avoid insulin-induced hypokalemia [46].

There are four main types of potassium-containing preparations: potassium chloride (KCl), potassium bicarbonate, potassium citrate, and potassium phosphate. Potassium phosphate solution is particularly useful when hypophosphatemia is associated, and citrate or bicarbonate solutions, when acidosis is installed [40]. In most situations, however, the solution of choice is potassium chloride. An adverse effect of oral KCl tablets (usually containing 8 mmol K^+^) is the irritation of the gastrointestinal tract mucosa, which can even lead to ulcerations or bleeding. For this reason, tablet ingestion must be accompanied by a large volume of fluid. The use of potassium-sparing diuretics during K^+^ replacement treatment can ease the onset of hyperkalemia, especially in diabetic patients with reduced GFR, using non-steroidal anti-inflammatory drugs, ACEi, or ARBs [44]. An interesting approach in diabetic patients prone to hypokalemia is to encourage the intake of potassium-rich foods, such as bananas, tomatoes, lentils, nuts, fish meat, etc., always keeping in mind the glycemic load of each item.

## 4. Conclusions

DM has a growing prevalence worldwide regardless of age, sex and ethnicity. Patients with this condition, if not well managed, are more susceptible to developing a series of other conditions, such as electrolyte disturbances, which can be lethal, especially in patients with other comorbidities, such as HF and CKD.

In this review, we presented the main reasons why diabetic patients are so vulnerable to developing hypokalemia, the mechanisms behind it, and the current methods of treatment and management of this potentially lethal condition.

Rapid identification of hypokalemia can prevent the occurrence of serious, life-threatening complications, such as cardiac arrhythmias and respiratory muscle impairment. Therefore, indications for urgent treatment include severe or symptomatic changes in K^+^ levels, electrocardiography changes, or the presence of certain comorbid conditions. Collectively, these data indicate that, as with all diagnostic aids, clinical correlation is indicated and K^+^ intake should be addressed.

## Figures and Tables

**Figure 1 medicina-58-00431-f001:**
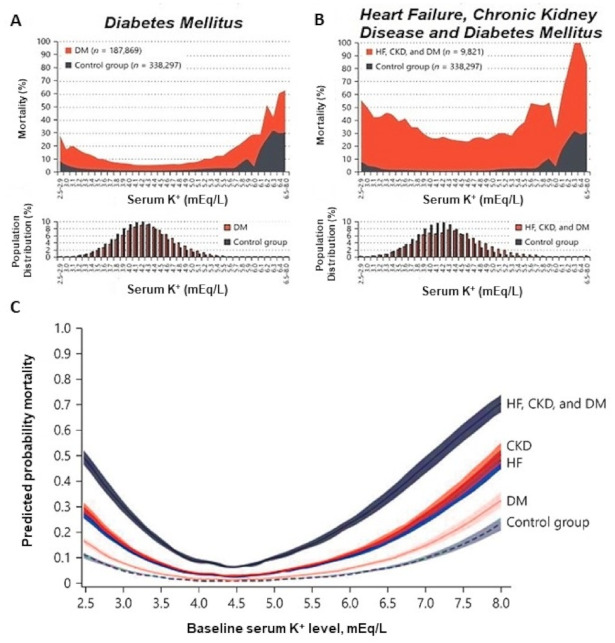
Mortality according to serum potassium concentration in individuals with diabetes mellitus, associated or not with other comorbidities and distribution of serum potassium concentration in these populations. (**A**) Diabetes Mellitus versus Control Group. (**B**) Heart Failure, Chronic Kidney Disease, and Diabetes Mellitus versus Control Group. (**C**) Correlation between serum potassium levels and predicted probability of mortality according to each comorbidity and when comorbidities were combined. CKD: Chronic Kidney Disease; DM: Diabetes Mellitus; HF: Heart Failure. Adapted from [2].

**Figure 2 medicina-58-00431-f002:**
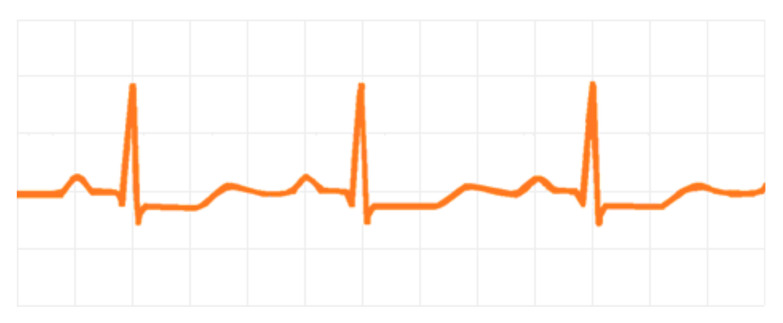
Drawing of an ECG showing the main changes during hypokalemia: Extension of QT interval, T wave flattening with ST-T depression, and U waves.

**Figure 3 medicina-58-00431-f003:**
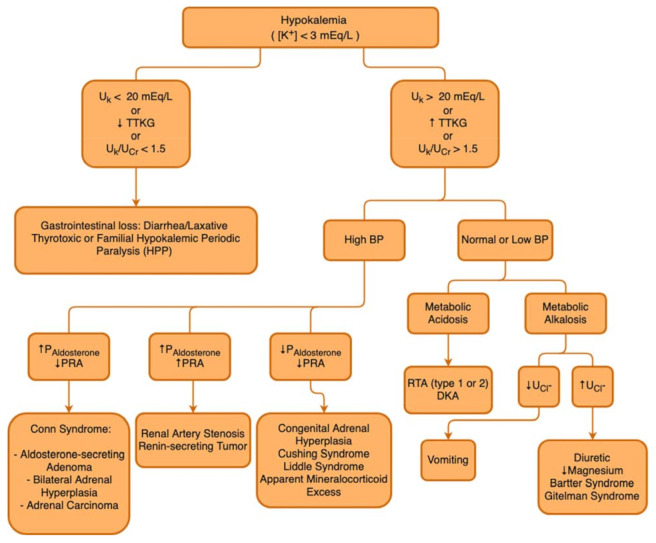
Hypokalemia diagnostic flowchart. *UK*: Urinary Potassium; TTKG: Transtubular Potassium Gradient; *UCr*: Urinary Creatinine; BP: Blood Pressure; DKA: Diabetic Ketoacidocis; *PAldosterone*: Plasmatic Aldosterone; PRA: Plasmatic Renin Activity; RTA: Renal Tubular Acidosis; *UCl*^−^: Urinary Chloride.

**Table 1 medicina-58-00431-t001:** Main mechanisms of hypokalemia.

Drug-Induced Transcellular Shifts	Induced Gastrointestinal Losses
Insulin Beta-2 agonists Verapamil Chloroquine Barium	Laxatives Enemas Prolonged vomiting Volume depletion
**Nondrug-Induced Transcellular Shifts**	**Drug-Induced Potassium Losses**
Neoplasms Thyrotoxicosis Primary hyperaldosteronism Familial hypokalemic paralysis Delirium tremens Barium intoxication Cushing syndrome	Thiazide diuretics Loop diuretics Glucocorticoids Mineralocorticoids Penicillin Aminoglycosides Amphotericin B Glycyrhizza glabra
**Nondrug-Induced Potassium Losses**	
Low dietary intake Diarrhea Metabolic alkalosis Type I and II renal tubular acidosis Hypomagnesemia Acute and chronic complications of DM Exocrine pancreatic insufficiency Infections Inflammatory bowel diseases Malabsorptive syndromes Bartter syndrome Gitelman syndrome Acute tubular injuries	

**Table 2 medicina-58-00431-t002:** Diagnostic criteria and typical deficits in the hyperglycemic hyperosmolar [13,36]. Na^+^: Sodium, Cl^−^: Chloride, HCO_3_^−^: Bicarbonate, K^+^: Potassium, PO_4_^−^: Phosphate, Mg^2+^: Magnesium, Ca^2+^: Calcium.

Diagnostic Criteria	HHS	DKA
pH	>7.30	≤7.30
Plasma Glucose	>540–600 mg/dL	>250 mg/dL
Serum Bicarbonate	>15–18 mEq/L	<18 mEq/L
Plasma and Urine Ketones	None or trace	Positive
Anion Gap: Na^+^ − (Cl^−^ + HCO_3_^−^)	<12	>12
Serum osmolality	>320 mOsm/Kg	Variable
Glycosuria	++	++
Typical Deficit		
Water (mL/Kg)	100–200 (9 L)	100 (6 L)
Na^+^ (mEq/Kg)	5–13	7–10
Cl^−^ (mEq/Kg)	5–15	3–5
K^+^ (mEq/Kg)	4–6	3–5
PO_4_^−^ (mmol/Kg)	3–7	5–7
Mg^2+^ and Ca^2+^	1–2	1–2

++ (glycosuria ∼30 mmol/L).

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
