# Peer review of "Hypokalemia in Diabetes Mellitus Setting"

_medicina, 2022, doi:10.3390/medicina58030431_

Round 1
Reviewer 1 Report
The manuscript entitled “Hypokalemia in diabetes mellitus setting” submitted by Ring et al. performed a review on hypokalemia in diabetic patients. The authors mentioned the different factors associated with hypokalemia in diabetic patients, diagnosis of hypokalemia and management of hypokalemia. The manuscript was well written and has merit for publication. However, some issues should be resolved from the authors’ side.
- There is an association of genetic mutation with hypokalemia in diabetic patients like the mutation in SLC12A3, CLDN10, HSD11B2, CLCNKB etc. Besides, some genetic disorders (Glucocorticoid-remediable hypertension, Bartter syndrome, Gitelman syndrome, Liddle syndrome, Gullner syndrome, Glucocorticoid receptor deficiency etc.) are connected with hypokalemia. Authors should include a section on the connection between genetic mutation and genetic disorder with hypokalemia, as some of these diseases also connect with diabetes.
- In line 25, the authors mentioned DKK as an abbreviated form of diabetic ketoacidosis, whereas, in the rest of the manuscript, authors use DKA as an abbreviated form of diabetic ketoacidosis.
- In the case of thiazide diuretics, an increased Na+ supply to the collecting duct causes hypokalemia (line 83-86), whereas, in the case of glucocorticoids, a reduced Na+ supply to the distal tubule and collecting duct causes hypokalemia (line 126-128). So how do these two opposite mechanisms cause hypokalemia?
- Authors should reduce plagiarism below or less than 20%, and the present status of plagiarism is 25% without references.
Author Response
We would like to thank the reviewers for their comments, which certainly contributed to strengthening our manuscript. We reviewed the manuscript accordingly and highlighted the changes using the TrackChanges tool throughout the manuscript.
- There is an association of genetic mutation with hypokalemia in diabetic patients like the mutation in SLC12A3, CLDN10, HSD11B2, CLCNKB etc. Besides, some genetic disorders (Glucocorticoid-remediable hypertension, Bartter syndrome, Gitelman syndrome, Liddle syndrome, Gullner syndrome, Glucocorticoid receptor deficiency etc.) are connected with hypokalemia. Authors should include a section on the connection between genetic mutation and genetic disorder with hypokalemia, as some of these diseases also connect with diabetes.
Response: Thank you for pointing this out. We created a new section regarding the genetic causes of hypokalemia and added the following information, starting at line 259 onwards.
Hypokalemia may be also a finding in a broad setting of genetic disorders. Briefly, from a pathophysiologic perspective, hypokalemia may be divided into disorders with a low urine K+ excretion or disorders with a high urine K+ excretion, as reviewed elsewhere.
In the genetic disorders associated with low urine K+ excretion rate, three mechanisms are involved: (a) increased K+ shift, including familial hypokalemia periodic paralysis or FPP (autosomal dominant disorder caused by mutations on ion channel genes encoding the dihydropyridine-sensitive voltage-gated Ca2+ channel α1-subunit (CACNA1S) [FPP type I] and tetrodotoxin-sensitive voltage-gated Na+ channel α-subunit [SCN4A] [FPP type II] of skeletal muscle) and Andersen-Tawil syndrome (associated with Mutations in the gene (KCNJ2) encoding a pore-forming subunit of the inward rectifier K+ channel protein, Kir2.1, which is expressed in heart and skeletal muscles); (b) defects in intestinal tract (characterized by a mutation in the downregulated in adenoma (DRA) gene encoding a Cl--OH- (HCO3-) exchanger expressed in the apical membranes of the colon and ileum, which leads to watery diarrhea, hypochloremic metabolic acidosis and hypokalemia); (c) defects in exocrine glands (cystic fibrosis is associated with a defective chloride reabsorption by the dysfunctional CFTR [cystic fibrosis transmembrane regulator] in the sweat ducts which is responsible for excessive Cl- and Na+ loss in sweat, leading to ECF volume depletion and, ultimately, to secondary hyperaldosteronism). To note, in familial hypokalemia periodic paralysis, attacks can be induced not only by rest after exercise, carbohydrate-rich meals, or exposure to cold but also by the administration of glucose or insulin or glucocorticoid, which can put diabetic patients at greater risk.
In the genetic disorders associated with high urine K+ excretion rate, two mechanisms are involved: (a) increased urine flow rate to cortical collecting ducts and (b) increased K+ concentration in the cortical collecting ducts. An increased urine flow rate to cortical collecting ducts is caused by increased excretion of electrolytes due to diuretic use or when a tubular defect is found or the increase in osmole excretion is due to non-electrolytes, such as mannitol, glucose, or urea. Therefore, a diabetic patient with higher levels of plasma glucose may present hypokalemia which is secondary to an increase in urine flow rate to cortical collecting ducts due to glycosuria.
When increased K+ concentration in the cortical collecting ducts is observed, we may speculate that a disorder characterized by fast Na+ reabsorption in the cortical collecting ducts is present. The most common disorders found in this setting are genetic hypokalemia associated with mineralocorticoid excess state, including glucocorticoid-remediable aldosteronism (a disorder caused by mutations in 11β-hydroxylase (CYP11B1) gene which encodes the aldosterone synthase gene for aldosterone biosynthesis in the adrenal zona glomerulosa and is regulated by angiotensin II and aldosterone synthase (CYP11B2) which encodes 11β-hydroxylase gene for cortisol biosynthesis in the adrenal zona fasciculate and is regulated by adrenocorticotrophic hormone [ACTH]), congenital adrenal hyperplasia due to 11β-hydroxylase or 17α-hydroxylase deficiencies [11β-OHD and 17α-OHD], Liddle´s syndrome (mutations in both β and γ subunits of ENaC associated with a deletion or alteration in their cytoplasmic C termini, which leads to a lack of its internalization via clathrin-coated pits pathway or its degraded via Nedd4 pathway, and therefore ENaC remains in an activated form on the cell surface), and apparent mineralocorticoid excess (a rare disorder that is caused by mutations in the gene (HSD11B2) encoding renal-specific 11β-hydroxysteroid dehydrogenase type 2 (11β-HSD2), which is responsible for converting cortisol to cortisone in the principal cells of distal tubules and crucial for protecting the mineralocorticoid receptor from being occupied by cortisol). In all these disorders, the excess of mineralocorticoid, either by augmented aldosterone secretion or production of other steroids, upregulates ENaC activity and leads to sodium reabsorption and hypertension associated with hypokalemia.
The second group of disorders associated with increased K+ concentration in the cortical collecting ducts includes a mechanism in which Cl- reabsorption in this segment is diminished, which includes disorders such as hypochloremic metabolic alkalosis, in particular Bartter´s syndrome (an autosomal recessive renal tubular disorder characterized by defective reabsorption of NaCl in the Henle´s loop, including five subtypes of mutations in genes encoding the Na+-K+-2Cl- cotransporter [NKCC2], K+ channel [ROMK], kidney-specific Cl- channel [CLCNKB], barttin [BSND] and calcium-sensing receptors [CaSR]) and Gitelman´s syndrome (secondary to defective reabsorption of NaCl in the distal convoluted tubule due to mutations in the SLC12A3 gene, which encodes the thiazide-sensitive Na+/Cl- cotransporter [NCC] on the apical membrane of that tubule). In both syndromes, hypokalemia is associated with renal K+ and Na+ wasting, with low or normal blood pressure and secondary hyperreninemia and hyperaldosteronism. Whereas in Bartter´s syndrome, a High urine Ca2+ and Mg2+ excretion is found, in Gitelman´s syndrome a low urine Ca2+ and high Mg2+ excretion is invariably found.
Importantly, Gitelman´s syndrome patients may be at greater risk of developing insulin resistance and type 2 DM [Ref], as chronic hypokalemia and hypomagnesemia impair insulin secretion and sensitivity, whereas hyperaldosteronism increases insulin resistance. From a molecular and cellular point of view, when glucose enters the pancreatic ß cell via GLUT2 transporter, it is metabolized to glucose-6-phosphate leading to changes in the intracellular concentration of adenine nucleotides that inhibit the KATP channel and cause its closure. When it occurs, the membrane depolarization activates subsequently the voltage-dependent calcium channels, leading to an increase in intracellular calcium, which triggers insulin exocytosis. Mutations in the ATP-Sensitive Potassium-Channel Subunit Kir6.2 lead to permanent neonatal DM [Ref].
In addition, hypomagnesemia found in Gitelman´s syndrome patients is implicated in a reduced tyrosine kinase activity at the insulin receptor level, and dysregulates K+ -ATP and L-type Ca2+ channels in the ß cells, which impairs insulin activity and secretion. Hyperaldosteronism can increase reactive oxygen species, accelerate endothelial remodeling, which can reduce the delivery of insulin for glucose metabolism, and promote insulin resistance by reducing insulin receptor substrate-1 expression, and by blocking the downstream protein kinase B signaling in the vascular smooth muscles [Ref].
The second cause of increased K+ concentration in the CCD comprises hyperchloremic metabolic acidosis, in particular, inherited isolated proximal RTA (renal tubular acidosis associated with defects in the electroneutral Na+-H+ exchanger 3 [NHE3] in the luminal membrane or the electrogenic Na+-HCO3- cotransporter [NBC1] in the basolateral membrane, which is associated with a reduction in the reabsorption of filtered HCO3-), inherited distal RTA (mutation in the genes encoding two H+-ATPase subunits specific to intercalated cells, including ATP6V1B1 and ATP6V0A, or mutations in H+-K+-ATPase and AE1 Cl--HCO3- exchanger [SLC4A1], leading to an impaired ability to excrete H+ in distal tubules), and inherited proximal and distal RTA (mutation in the gene encoding carbonic anhydrase II which generates H+ in distal convoluted tubules) [Ref].
In proximal RTA, the initial insult is associated with an excess of HCO3- excretion and the urine pH is higher than 6.5. However, over time HCO3- serum levels drop and an impaired absorption is again sufficient to acidify urine to lower than 5.5. Thus, proximal HCO3- excess loss in the urine causes an increase in urine flow rate to the distal nephron, leading to potassium wasting and activation of RAAS from mild hypovolemia and, therefore, to hypokalemia. Other features associated with RTA include the augment in glucose, uric acid, phosphate, and amino acids in urine (Fanconi syndrome) [Ref].
In distal RTA, the decreased H+ in tubule lumen draws out K+ causing hypokalemia. In addition, these patients present calcium phosphate kidney stones caused by decreased citrate excretion and hypercalciuria, as well as by the fact that salts are more likely to precipitate at higher urine pH [Ref].
- In line 25, the authors mentioned DKK as an abbreviated form of diabetic ketoacidosis, whereas, in the rest of the manuscript, authors use DKA as an abbreviated form of diabetic ketoacidosis.
Response: We correct the wrong abbreviation.
- In the case of thiazide diuretics, an increased Na+ supply to the collecting duct causes hypokalemia (line 83-86), whereas, in the case of glucocorticoids, a reduced Na+ supply to the distal tubule and collecting duct causes hypokalemia (line 126-128). So how do these two opposite mechanisms cause hypokalemia?
Response: Thank you for bringing this topic to discussion.
We added the following information about the use of diuretics (line 114): “In the case of the diuretics (loop diuretics and thiazide diuretics), the increased amount of intra-luminal Na+ delivered to the distal segments of the nephrons enters the principal cells via ENaC channel. This creates a negative electrical charge inside the lumen. Activation of Na+-K+-ATPase causes it to pump Na+ into the blood and exchange for K+. Drawn by the negative electrical charge in the lumen, K+ will leave the cell to enter the lumen via the ROMK channel, leading to hypokalemia.
Inside the intercalated cell of the collecting ducts, the stimulation of H+-K+-ATPase pump, which along with the negative charge in the collecting duct lumen, causes H+ to exit intercalated cells into the lumen. Therefore, alkalosis metabolic develops (serum HCO3- concentration, 28-36 mmol/L) and leads to hypokalemia as well [4].
We corrected the information regarding the use of drugs with mineralocorticoid or glucocorticoid effects (line 123), as the mechanism of action is similar to the use of loop diuretics and thiazide diuretics, as aldosterone stimulates ENaC, ROMK, and Na+-K+-ATPase. Importantly, glucocorticoids are usually given at doses that produce minimal mineralocorticoid stimulation (comparing to cortisol, prednisone and prednisolone have 0.8 mineralocorticoid potency, whereas hydrocortisone mineralocorticoid potency is equal to cortisol, fludrocortisone has 125-150 potency mineralocorticoid when compared to cortisol, and dexamethasone does not have mineralocorticoid potency) to avoid the side effects associated with activation of aldosterone pathway, which leads to hypokalemia, volume expansion, and hypertension.
- Authors should reduce plagiarism below or less than 20%, and the present status of plagiarism is 25% without references.
Response: We apologized for the rate of similarity that was detected. We reviewed the manuscript carefully (we rewrote several paragraphs) and after a new evaluation of the Turnitin, the similarity is acceptable.
Reviewer 2 Report
The current manuscript is well-written and focuses on the causes and effect of hypokalemia on diabetes. However, there are some important issues to clarify.
First, I think it would be better discuss the role of potassium level in prediabetes, as the previous studies show that the inverse relationship between potassium and diabetes (
Second, please organize and summarize the medications and states which causes hypokalemia in one table.
Third, please describe the effects of prolonged hypolakemia in diabetes. I think it would better to reduce the contents on DKA/HHS and management of hypokalemia since the manuscript is not a guideline, but a review.
Minor points.
1st, please describe the original form of figure 1 in detail.
2nd, please describe the medications causes hypokalemia in the paragraph 2.1.
Author Response
We would like to thank the reviewers for their comments, which certainly contributed to strengthening our manuscript. We reviewed the manuscript accordingly and highlighted the changes using the TrackChanges tool throughout the manuscript.
The current manuscript is well-written and focuses on the causes and effect of hypokalemia on diabetes. However, there are some important issues to clarify.
- First, I think it would be better discuss the role of potassium level in prediabetes, as the previous studies show that the inverse relationship between potassium and diabetes (Arch Intern Med. 2010;170(19):1745-1751, Hypertension. 2006; 48(2):219–224).
Response: Thank you for the comments. We added the following information in line 39 onwards:
There is also an association between hypokalemia and the risk of developing diabetes in prediabetic individuals [Ref]. There is evidence of an inverse relationship between serum K+ level and the risk of developing incidental DM, regardless of diuretic use. One study found hazard ratios of 1.2-1.3 for the development of incidental DM in subjects with serum potassium K+ below 5.0 mEq/L when compared to subjects whose potassium levels were above this range [Ref]. Hypokalemia induced by the use of thiazide diuretics was also associated with an increased risk of developing DM. However, early-stage control of thiazide-induced hypokalemia may correct glucose intolerance to prevent the development of diabetes in these individuals [Ref]. Despite plasma insulin and serum glucose increase and serum K+ decreases in hypertensive patients under hydrochlorothiazide monotherapy, no changes of serum glucose were noted whether serum K+ levels were greater than 4.0 mEq/L or below 4.0 mEq/L [Ref].
The relation of hypokalemia and hyperglycemia is based on the function of the ATP-sensitive potassium (KATP) channel in islet cells [Ref]. Thus, when glucose enters ß-cell through GLUT2 transporter it is metabolized to glucose-6-phosphate. The increase in intracellular adenine nucleotides inhibits KATP and leads to its closure. The subsequent membrane depolarization activates the voltage-dependent Ca2+ channels, which triggers insulin exocytosis after the increase in Ca2+ intracellular. Thus, hypokalemia is associated with the development of hyperglycemia through the impairment of potassium-dependent insulin release in response to glucose overload.
Second, please organize and summarize the medications and states which causes hypokalemia in one table.
Response: Thank you for your suggestion. We summarized the main mechanisms of hypokalemia in one single table (Table 1). Please verify line 51.
- Third, please describe the effects of prolonged hypolakemia in diabetes. I think it would better to reduce the contents on DKA/HHS and management of hypokalemia since the manuscript is not a guideline, but a review.
Response: Thank you for pointing this out. We added the following information in line 384 onwards:
Although chronic or persistent hypokalemia may be asymptomatic in some individuals, patients with DM may have this condition worsened by diarrhea or vomiting, which can occur during acute complications of DM [Ref]. Nocturia and polyuria can also be exacerbated, especially in individuals predisposed to persistent hypokalemia, as in Bartter and Gitelman syndromes. Hypokalemia-induced polyuria is related to an impairment of vasopressin action in collecting ducts. In addition, insulin treatment can also promote K+ shift into cells. Therefore, hypokalemia can have worse consequences in diabetic patients, which puts these individuals at greater risk of chronic hypokalemia. In this group of patients, cardiovascular diseases are found more often, making them more vulnerable to cardiac arrhythmias, fluid depletion, and worsening neuropathy from muscle weakness [Ref, Ref ]. To note, KATP channels may not function properly in the DM setting because its expression is reduced in myocardium cells and aortic smooth muscle cells, resulting in impaired heart and vascular function [Ref]. Consequently, hypokalemia may affect the membrane potential and pose a decreased response to stress conditions, such as hypoxia and oxidative stress.
Importantly, as hypokalemia may lead to hyperglycemia due to the impairment of insulin secretion and peripheral glucose utilization, a vicious circle is triggered where hypokalemia worsens glucose control and vice-versa.
Minor points
- 1st, please describe the original form of figure 1 in detail.
Response: As suggested, we improved the description of Figure 1. Mortality according to serum potassium concentration in individuals with diabetes mellitus, associated or not with other comorbidities and distribution of serum potassium concentration in these populations. (a) Diabetes Mellitus versus Control Group. (b) Heart Failure, Chronic Kidney Disease and Diabetes Mellitus versus Control Group. (c) Correlation between serum potassium levels and predicted probability of mortality according to each comorbidity and when comorbidities were combined. CKD: Chronic Kidney Disease; DM: Diabetes Mellitus; HF: Heart Failure.
We also added in line 30 the following information: Therefore, there is an association between baseline serum K+ levels and mortality in these patients (Figure 1). All three conditions (CKD, HF, and DM) can be associated with higher mortality rates over 18-month follow-up when compared with control individuals. All-cause mortality was also higher at the extremes, both in K+ values below 4.0 mEq/L and in values above 6.0 mEq/L [2].
- 2nd, please describe the medications causes hypokalemia in the paragraph 2.1.
Response: We added the following information in line 51: Other medications that contribute to hypokalemia in individuals with diabetes are insulin, beta-2 agonists (through activation of the Na+-K+-ATPase pump), antiarrhythmic agents, glucocorticoids and mineralocorticoids, antibiotics (penicillin and aminoglycosides), antifungals (amphotericin B), and excessive use of laxatives.
Round 2
Reviewer 2 Report
It would be better to remove table 2 and figure 3. Because this is not a guideline, but a review.
Author Response
Reviewer 2:
It would be better to remove table 2 and figure 3. Because this is not a guideline, but a review.
Response: Thank you for your suggestion. Table 2 and figure 3 were removed, as recommended.